# The Relative Age Effect in the Best Track and Field Athletes Aged 10 to 15 Years Old

**DOI:** 10.3390/sports10070101

**Published:** 2022-06-28

**Authors:** Eduard Bezuglov, Maria Shoshorina, Anton Emanov, Nadezhda Semenyuk, Larisa Shagiakhmetova, Alexandr Cherkashin, Bekzhan Pirmakhanov, Ryland Morgans

**Affiliations:** 1Department of Sports Medicine and Medical Rehabilitation, Sechenov First Moscow State Medical University of the Ministry of Health of the Russia Federation, 119991 Moscow, Russia; e.n.bezuglov@gmail.com (E.B.); kaisough@yandex.ru (M.S.); rylandmorgans@me.com (R.M.); 2“Smart Recovery” Sports Medicine Clinic LLC, 121552 Moscow, Russia; 5184436@gmail.com (A.E.); nadya480@yandex.ru (N.S.); larisas777@mail.ru (L.S.); 3High Performance Sport Laboratory, Moscow Witte University, 300028 Moscow, Russia; 4Sirius University of Science and Technology, 354349 Sochi, Russia; 5Russian Athletics Federation, 119991 Moscow, Russia; alexandercherkashin@yahoo.com; 6Department of Epidemiology, Biostatistics and Evidence-Based Medicine, Faculty of Medicine and Health Care, Al-Farabi Kazakh National University, Almaty 050040, Kazakhstan; 7FC Kairat, Almaty 050054, Kazakhstan

**Keywords:** young athletes, track and field, relative age effect

## Abstract

(1) The purpose of this study was to examine the prevalence of the relative age effect (RAE) in the best young (10 to 15 years old) track and field athletes. (2) Hypothesis: The prevalence of the RAE in the best young track and field athletes of both genders will be evident in all age groups from 10 to 15 years old, which may be associated with the significant relationship between biological maturity, chronological age, and the development of physical qualities. (3) Materials and methods: In total, 1778 athletes volunteered for this study. The sample was based on the results of the best young athletes who participated in the final tournaments of the national competition “Shipovka Yunykh”(“Running spikes for young athletes”), which have been held since 1981. The sample group consisted of male and female athletes classified into specific age groups: 10 to 11 years old (*n* = 579), 12 to 13 years old (*n* = 600), and 14 to 15 years old (*n* = 599). Analysis was performed using Jamovi 1.8.1. The Chi-square test was used to compare the RAE between different groups. (4) Results: A wide distribution of the RAE was revealed both in the general sample and in boys and girls. The percentage of “early-born” athletes was 37.6% while only 12.3% were “late-born” athletes. **The difference in the severity of the RAE may reflect the small sample of athletes from the fourth quartile, which was significantly less than the sample of boys from the fourth quartile (*p* = 0.04, OR 1.68, 95% CI 1.02–2.78).** The RAE was also evident in all age groups of boys and girls, without any statistically significant differences in the severity (*p* > 0.05, Chi = 2.135, **V = 0.02**). In the 14- to 15-year-old male athletes group, the number of early-born compared to late-born athletes peaked. **The RAE was most common amongst the most successful track and field athletes**. Among the competition medalists during the analyzed time period, more than 50% of athletes were born in the first quarter and no athletes were born in the fourth quarter.

## 1. Introduction

During recent decades, many studies have appeared that have described the RAE in various sports [1,2]. The RAE refers to the over-representation of athletes born closer to the criteria used to distinguish between different athlete age group categories [3]. In the elite sporting world, most often, the calendar starting point is January 1st. This is the most commonly applied method used by leading sports academies and organizers to divide athletes into different age groups when determining participation in major competitions. Thus, sports with a wide distribution of the RAE, such as football, hockey, and basketball, describe the over-representation of athletes born between January and March and the under-representation of athletes born between October and November [4,5,6].

The RAE has previously been reported in a range of athletes from different age categories and genders, which has been suggested may lead to lower overall performance levels and, for some athletes, discontinuation before reaching their full potential [7]. In a recent study [8], found that in 39,590 athletes from the International Athletics Association’s Top-100 U18 category, early-born athletes were 5 times more likely to be included in this ranking list when compared to late-born athletes. Furthermore, the RAE was found to be more significant in male athletes [8,9] further reported that the RAE in elite U15-17 Spanish athletes was widespread, especially in the U15 group; therefore, this supports the “maturation-selection” notion as a mechanism of the RAE. In contrast [10], cited the first experience with official U13 athletic competitions was most notable in UK athletes while gender, event, and skill level also influenced the presence and magnitude of the RAE in different age categories.

The common practice of the RAE has been described in the youngest age groups (6 to 12 years old) in a variety of sports, including hockey, football, and alpine skiing [11,12,13,14,15]. At this age, it is possible that chronologically older children have both physical and cognitive advantages. In older age group athletes (15 to 18 years old) in these sports, it is widely reported that the RAE is most significant in elite male athletes [1,16]. The prevalence of the RAE in elite adolescent athletes in these sports may be due to chronologically older peers that are more likely to reach puberty earlier and, therefore, acquire advantages in strength, speed, endurance, and bone strength due to the influence of testosterone [17]. Thus, early-maturing late-born children may not be excluded from highly competitive sports, as some authors have noted, as chronological age increases, the number of late-maturing children significantly decreases [18,19].

In team sports such as football and hockey, the widespread acceptance of the RAE in adolescence is due to the very early selection processes for leading football and hockey organizations. Very often, this selection occurs at 6 to 8 years old, which leads to the initial over-representation of early-born children, which can persist until late puberty. However, in athletics, the primary selection process occurs during the first competition phase, normally at 13 years old, that is, before puberty.

Given that in youth athletic competitions all disciplines have a significant emphasis on strength and speed (sprint, long jumps), which are directly related to testosterone levels (and, therefore, biological maturity), it can be assumed that in athletics, the prevalence of the RAE will be most notable in younger age categories and in the most competitive groups, which creates discrimination against late-born late-maturing children.

In the contemporary scientific literature, there is scant research investigating the prevalence of the RAE in elite youth athletic groups. This study aimed to examine the prevalence of the RAE concept in the best young (10 to 15 years old) Russian athletes and, furthermore, recommend methods to possibly reduce bias in identifying future talent.

The main hypothesis of this study was that the RAE would be widespread in all athletes of both sexes and particularly in boys over 12 years old. A possible reason is the dramatic increase in testosterone levels during puberty, which may provide a chronological advantage for older adolescents, who are likely to be more biologically mature than their younger counterparts.

## 2. Materials and Methods

The study sample consisted of 1778 **young** (10 to 15 years old) athletes. The sample athletes volunteered and were based on the results of the best young athletes who competed in the final tournaments of the national competition “Shipovka Yunykh” (“Running spikes for young athletes”), which have been held since 1981.

The winners of regional qualifying tournaments, which are held a few months before “Shipovka Yunykh”, participated in these competitions. These competitions are held annually, and the winners are determined by the sum of three athletic disciplines. For all age groups, the disciplines are the 60 m sprint, the 600 m run, and the long jump. These competitions are held in stadiums that meet international standards, and the running results are determined by an electronic timing system. The best athletes were those ranked in the Top-100 from the all-around competitions.

The sample group consisted of male and female athletes classified into age groups: 10 to 11 years old (*n* = 579), 12 to 13 years old (*n* = 600), and 14 to 15 years old (*n* = 599). In 2017 (*n* = 578), 2018 (*n* = 600), and 2019 (*n* = 600), athletes participated in the final tournaments of the most prestigious national competitions. The number of participants in each age group varied from 579 to 600 in different years and this was determined by athletes who competed in all disciplines from the all-around competition. Ethics was approved by the local Ethics Committee of Sechenov University (N 06-21 dated 4 July 2021). To ensure confidentiality, all data were anonymized before analysis.

Based on the official competition protocols provided by the National Athletics Federation, two independent experts analyzed the date of birth data from the best young athletes in Russia. Competitions are traditionally held in three different age groups where each age group is classified by 1 January to 31 December for two consecutive years. These competitions are considered all-round, as events include the 60 m sprint and 600 m run, the long jump, and throwing of a “rocket”-type projectile.

All athletes were divided into four quartiles according to the month of birth: January to March as the first quartile (Q1, “early-born”); April to June as the second quartile (Q2); July to September as the third quartile (Q3); and October to December as the fourth quartile (Q4, “late-born”). Additionally, the prevalence of RAE among the top 3, top 5, and top 10 young track and field athletes in their age groups was determined. This quarterly division is most commonly used in studies examining the prevalence of the RAE in sports, including track and field [5,20,21]. The prevalence of the RAE was determined in young athletes born in different quarters in the general sample, in different age groups, and separately for boys and girls.

## 3. Statistical Analysis

Data were stored in MS Excel. Analysis was performed using The jamovi project (2021), jamovi (Version 1.8.1) [Computer Software]. Retrieved from https://www.jamovi.orgc. The absolute and relative numbers of athletes were calculated. The Chi-square test was used to compare the effect of the RAE between different groups. Results were considered statistically significant at *p* < 0.05. **The odds ratio and 95% CIs were calculated for the 2 × 2 tables. For tables larger than 2 × 2, Cramer’s V was calculated.**

## 4. Results

In this study, the dates of birth of 1778 young track and field athletes (aged 10 to 15 years old) who participated in the “Shipovka Yunykh” track and field competition in 2017–2019 were analyzed. There were 879 boys and 899 girls. The number of athletes from both sexes in different age groups is shown in Table 1.

The representation of young athletes born in different quartiles was analyzed. A wide distribution of the RAE was revealed both in the general sample and in boys and girls. The percentage of “early-born” athletes was 37.7% while only 12.3% were “late-born” athletes (see Figure 1).

The RAE for all age group boys and girls showed no statistically significant differences (Χ^2^ = 2.135, *p* = 0.54, V = 0.02) (see Figure 2).

When examining the prevalence of the RAE in different age groups and in boys and girls from different age groups, it was found that the RAE was evident in all three age groups without statistically significant differences in the severity of the RAE between them (Χ^2^ = 10.576, *p* = 0.10, V = 0.031). The prevalence of the RAE between boys and girls in each age group showed statistically significant differences in the prevalence of the RAE in boys and girls in the 14- to 15-year-old age groups. In this group, the RAE in boys was more significant than in girls (Χ^2^ = 8.16, *p* = 0.043, V = 0.067). The difference in the severity of the RAE was due to the small sample of athletes in the fourth quartile, which was significantly less than the sample of boys in the fourth quartile (*p* = 0.04, OR 1.68, 95% CI 1.02–2.78). No differences were found between the other quartiles in this group. There were no statistically significant differences in the severity of the RAE **between boys and girls** in the age groups 10 to 11 years old (Χ^2^ = 1.61, *p* = 0.66, V = 0.03) and 12 to 13 years old (Χ^2^ = 3.04, *p* = 0.38, V = 0.041) (Figure 3, Figure 4 and Figure 5).

An additional analysis of the birth dates of the most successful young track and field athletes of both sexes (top 3, top 5, and top 10 in each age group) provided evidence of an extremely strong prevalence of RAE, which was directly linked to the success of the athletes’ performance: no athletes born in the fourth quarter were among the most successful athletes (top 3), and the number of such athletes of both sexes in the top 5 and top 10 was significantly lower than in the whole sample studied (Figure 6, Figure 7 and Figure 8).

## 5. Discussion

This original study was the first of its kind to obtain data on the prevalence of the RAE in a sample group of the best Russian athletes aged 10 to 15 years old, and among the most successful track and field athletes of both sexes (top 3, top 5, and top 10). This examination provided evidence that the RAE was significant in a sample of the **best young athletes consisting of boys and girls, with the greatest expression of the RAE in boys aged 14 to 15 years old**. **Thus, this study’s hypothesis was confirmed.** These data are consistent with previously reported results in young athletes of various competitive levels [9,10]. **According to the current literature, the key aspects underlying the RAE phenomenon are personality traits, the athletes’ development environment, and the type of sport.**

The main role in creating and maintaining the RAE can be determined by various social agents, including parents, coaches, and athletes themselves, through the effects of Matthew, Pygmalion, and Galatea, respectively, and their combined impact [22]. Additionally, a model based on individual constraints, task constraints, and environmental constraints to explain the RAE has been proposed [23]. A possible noteworthy factor of the common occurrence of the RAE in elite successful young athletes is the relationship between selection and the status of maturation (selection–maturation hypothesis) [5]. Thus, maturation status can be attributed to the task constraints previously described [23]. A further explanation for this concept in athletics, a sport where strength and speed are fundamental in most disciplines in early adolescence and advanced technical equipment is not required for success, is that early-born and thus early-maturing athletes have a significant advantage due to the physical nature of the sporting demands. For example [24], found that in 7500 young athletes from the Swiss Talent Identification Program, a significant difference (up to 10%) was shown in the 60 m sprint performance between early-born and late-born athletes aged 8 to 15 years old [24].

This difference in athletic performance may, among other factors, be related to changes in muscle metabolism associated with age and maturation [25]. Additionally, the emphasis on the age criteria of such a vitally important parameter for the development of strength and speed is the concentration of phosphocreatine, which was previously proven in boys aged 11 to 15 years old. The available data suggest that in most cases, in the highly competitive sporting communities, selection criteria are based on the degree of biological maturity in a specific time period (maturity-associated selection), thus creating the prerequisites for potential discrimination between late-born and late-maturing athletes. With this approach, in the pre-puberty period and during its onset (9 to 15 years old), early-born and early-maturing adolescent athletes can benefit, the results of which continue to gradually improve as they grow and mature, and the corresponding aerobic and anaerobic training and strength training further increases productivity [26,27].

An important characteristic of adolescent athletic competitions is the classification of age categories in an interval of two years, and not a singular year, such as in football and hockey. This creates a greater advantage with even more significance regarding the prerequisites for the success of early-born athletes and the potential elimination of late-born athletes. Thus, it is often the winners of a youth competition with a higher relative age that are more likely to be identified as talented because of the greater physical advantage they have over their “younger”, less mature peers [28]. These athletes are often then targeted by top coaches and given the opportunity to train in better conditions, which can further increase the differences in adolescent athletic performance. This may be most applicable to the most successful athletes who are ranked the highest. This study showed that the RAE was most pronounced among these athletes, and there were no athletes born in the fourth quarter among the medalists of these competitions (top 3) during the analyzed period.

The data in our study also showed that the most significant predominance of early-born athletes over late-born athletes was in male athletes aged 14 to 15 years old. It was during this specific chronological period that there was a significant increase in sporting success in young men, which may be primarily associated with an increase in the concentration of testosterone (an increase in adult males of up to 30 times compared to pre-puberty) [17]. Thus, there is a large difference between genders in circulating testosterone concentrations and a reproducible dose–response between circulating testosterone and muscle mass and strength, and circulating hemoglobin. Therefore, clearly, an increase in the testosterone concentration is directly related to the level of biological maturation and, therefore, can provide an advantage to early-maturing adolescents, especially during the peak growth spurt that occurs between 13 and 15 years [29]. In the analyzed “Shipovka Yunykh” competition data, in one age group, there were age differences greater than 20 months, contributing to the very low probability of the successful performance of chronologically younger athletes. This finding is noteworthy and competition organizers should perhaps group athletes according to birth dates in the same calendar year.

In our study, several limitations were identified. Primarily, we examined the prevalence of the RAE in the best young track and field athletes based on all-round results rather than individual disciplines. It is likely that the prevalence of the RAE in the best sprinters and 600 m runners would differ. The factors determining success in disciplines where strength and speed (sprinting) are the key qualities are not identical to those determining success in disciplines where endurance (600 m) is vital for success. Another limitation was that the RAE was examined in a relatively short period of time without considering the results of competitions held several decades ago. Given the global socio-cultural changes that took place in Russia in the 1990s, it can be assumed that the factors influencing the prevalence of the RAE may have changed. Unfortunately, the analysis of the participants’ dates of birth in “Shipovka Yunykh” in the USSR is impossible due to the lack of competition protocols. Additionally, the data obtained in our study revealed that a small number of late-born athletes were successful young track and field athletes. Thus, it may be of real practical interest in future studies to investigate the factors that allowed them to successfully compete with their older peers. It is possible that the majority of late-born young athletes mature early. Examining the prevalence of the RAE in junior athletes from both genders and specific disciplines and the age of onset of gender differences in athletic performance may also be a topic of future research.

It is important to note that despite the large number of publications devoted to investigating the RAE in sports, very few practical solutions have been proposed. Simple solutions such as increasing coaches’ awareness of the RAE presence and interpreting the results of physical and cognitive skills tests may be beneficial. Finally, considering the biological maturity status may probably also reduce the “drop-out” of late-born and (or) late-maturing young athletes during the primary stage of selection for elite sporting organizations. It should be reiterated that chronological age is not directly related to biological maturity status and, therefore, future studies involving athletes should note that the relationship between the two has yet to be identified.

## 6. Conclusions

In the best young athletes aged 10 to 15 years old examined, the RAE was widespread in both boys and girls. In male athletes from the 14- to 15-year-old age group, the number of early-born compared to late-born athletes peaked. The most pronounced RAE is expressed among the most successful young athletes, suggesting that early-born athletes have fewer opportunities to enter top sports organizations and access the best coaches, which increases their risk of leaving the sport as early during adolescence. These findings may be useful for specialist practitioners when assessing the potential talent of young athletes and for competition organizers when determining the selection criteria for age-specific categories.

The authors certify that there is no conflict of interest with any financial organization regarding the material discussed in the manuscript. The authors report no involvement in the research by a sponsor that could have influenced the outcome of this work.

## Figures and Tables

**Figure 1 sports-10-00101-f001:**
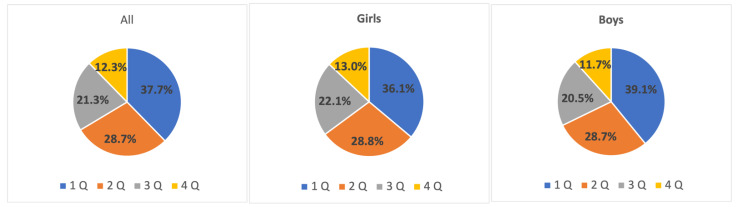
The prevalence of the RAE in the best young athletes **from both sexes** aged 10 to 15 years old.

**Figure 2 sports-10-00101-f002:**
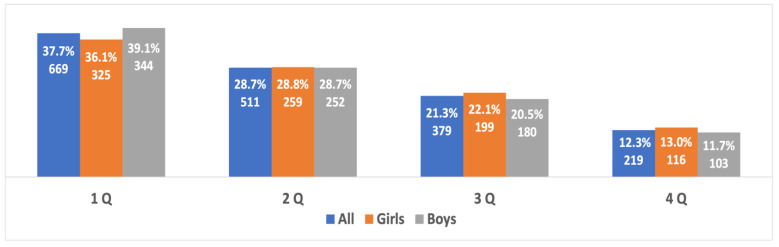
The prevalence of the RAE in the best young athletes aged 10 to 15 years old.

**Figure 3 sports-10-00101-f003:**
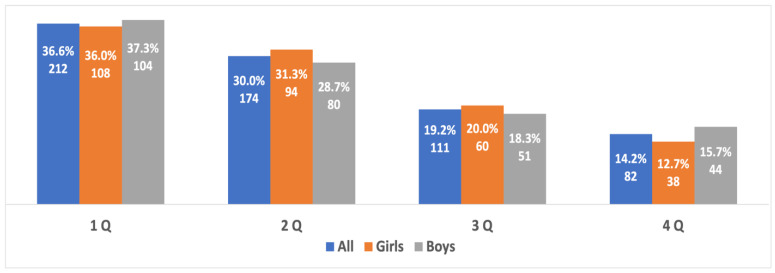
The RAE in the best young athletes from both genders aged 10 to 11 years old.

**Figure 4 sports-10-00101-f004:**
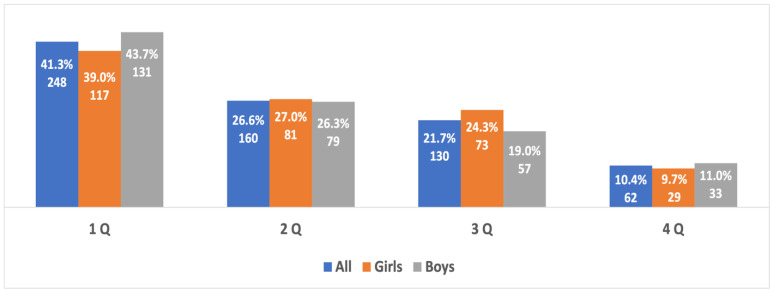
The RAE in the best young athletes from both genders aged 12 to 13 years old.

**Figure 5 sports-10-00101-f005:**
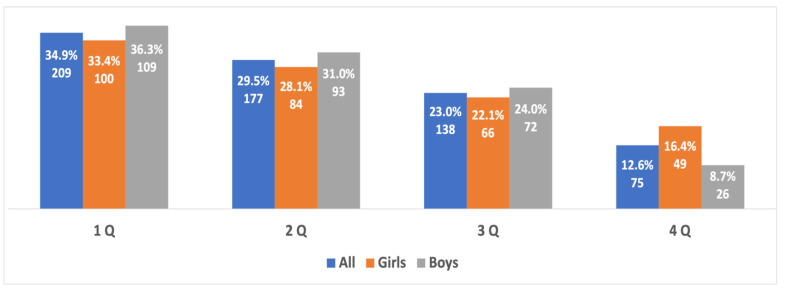
The RAE in the best young athletes from both genders aged 14 to 15 years old.

**Figure 6 sports-10-00101-f006:**
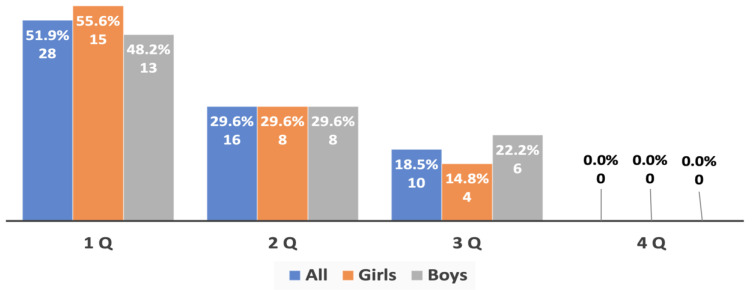
The RAE in the young athletes (top 3) from both genders aged 10 to 15 years old.

**Figure 7 sports-10-00101-f007:**
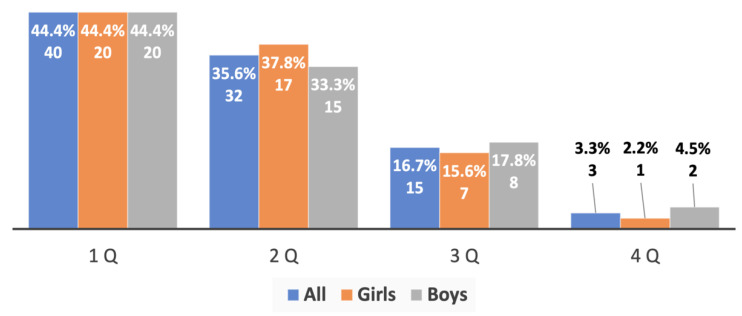
The RAE in the young athletes (top 5) from both genders aged 10 to 15 years old.

**Figure 8 sports-10-00101-f008:**
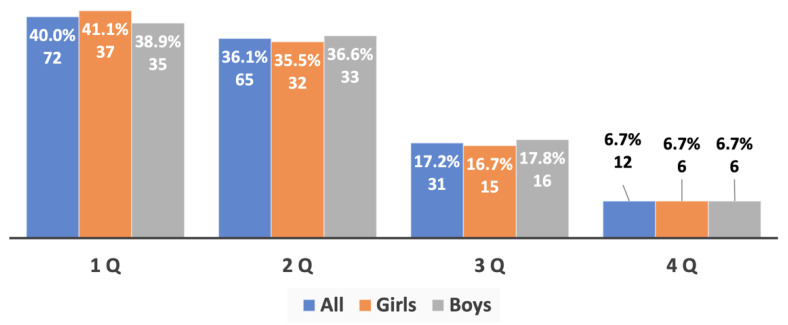
The RAE in the young athletes (top 10) from both genders aged 10 to 15 years old.

**Table 1 sports-10-00101-t001:** The number of young athletes from both sexes in different age groups.

Age Groups	Girls	Boys	All
10–11 years old	300	279	579
12–13 years old	300	300	600
14–15 years old	299	300	599
10–15 years old	899	879	1778

## Data Availability

The raw data supporting the conclusions of this article will be made available by the authors, without undue reservation.

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
