# Peer review of "The Relative Age Effect in the Best Track and Field Athletes Aged 10 to 15 Years Old"

_sports, 2022, doi:10.3390/sports10070101_

Round 1
Reviewer 1 Report
The present study examined the prevalence of the Relative Age Effect (RAE) in the best young (10 to 15 years) track and field athletes.
Despite the interesting topic, I strongly suggest following the comments to improve the quality of the manuscript.
Introduction
- Authors should include all hypotheses tested.
Materials and methods
2. Authors should include how the selection process of participants was really done.
3. L 98-99. This sentence was already mentioned in L 88 and 89.
4. L 100. Which type of competition protocols? Please give some examples.
5. L 106-111. All these sentences must be reinforced with scientifically validated references.
6. Statistical procedures might need to be discussed using a within-subjects approach since basic group comparisons were performed. It is not clear, the inferential test that was used to compare the difference between groups. Only chi-square analyses were used? How was this comparison attempted? Did the authors pool data for the comparison? How many data points were paired?
7. The authors should add the magnitude of differences between age and sex, to better understand the Δ. And please, discuss all the magnitudes found.
8. Results, discussion, and conclusion section should be rewritten accordingly to the previous commentaries.
9. It is not clear the limitation and practical application of this study. Please add and improve significantly.
10. Authors should do an effort to improve and add more illustrative figures. As it stands, it is very limited to highlight your findings.
Author Response
Dear Reviewer,
Thank you very much for your detailed reviewing our paper. We know that finally it increases its quality and we are very happy that you spend so much time to help us to improve it. Thank you for you specific and professional suggestion.
All the changes we mark with red color in the text.
We tried to fully reflect the answers to your suggestions in the text.
Now text looks much better and we hope that it fulfils your expectations.
Authors

Reviewer 2 Report
Dear authors,
I have completed the review of your manuscript entitled "The relative age effect among the best track and field athletes aged 10–15 years".
I found the manuscript interesting, but also found some aspects that can be improved. See below my detailed report. Also, I would suggest a linguistic revision-editing. I invite authors to consider suggestion #5. This would add relevant information and would also enhance the + value of having data from 40 years of youth sport competition.
- Language revision-editing of the article.
- L85: specify if track and field is considered a "major sport" in Russia (and Soviet Union berfore the early nineties.
- L116: What are natioanl birth rates in Russia ? Are RAE numbers congruent with the nation birthrates ?
- L126-L136: Moe clarity is needed here.
- I recommend authors to analysze RAE pattersn through the years (from 1981 to 1991: Soviet Union's regime and compare with 1990s to 2010s as the Russian system.
- Discussion: Needs to be improved (at leas I think it can): For example: Relate results with mechanisms that can be related with the nation's sport system. Most explanations related to physiological and growth-development hypotheses. I think it can be improved with more depth (comparing sports, etc...).
- References: if abbreviated journal titles, see reference #27 (and make the correction).
Author Response
Dear Reviewer,
Thank you very much for your detailed reviewing our paper. We know that finally it increases its quality and we are very happy that you spend so much time to help us to improve it. Thank you for you specific and professional suggestion.
All the changes we mark with red color in the text.
- According to the data of the federal fertility authority, there are no statistically significant differences.
5.
Yes, we also thought about it, but there are no correct protocols in the period from 1980 to 2000 at the disposal of the federations
6.
Thank you for this very interesting question.
We also believe that a number of socio-cultural factors underlie the widespread use of RAE among young athletes.
The main one is the direct dependence of the financial well-being of a coach on the results of his athletes in a specific period of time, and not for several years ahead.
This leads to the fact that more physically weak children are not interesting for coaches in Russia.
Another important nuance is the low awareness of Russian coaches about modern methods for determining skeletal age and the lack of modern equipment for testing physical qualities.
For example, 99% of coaches in Russia still use only a stopwatch in their work and do not have access to electronic systems for determining speed and strength.
We are ready to include this data in the article, but unfortunately we do not have relevant links for this.
If you have the desire and opportunity, then we are ready to carry out the following research on RAE in Soviet and Russian sports together.
We tried to fully reflect the answers to your other suggestions in the text.
Now text looks much better and we hope that it fulfils your expectations.
Authors

Reviewer 3 Report
The manusscript "The relative age effect among the best track and field athletes aged 10–15 years" analyses the prevalence of the Relative Age Effect (RAE) in the best young (10 to 15 years) track and field athletes. The analysis of track and field and dataset are intersting, however in the current version I have several major concerns.
1) The composition of the study group needs to be described in more detail. How many women and men participated in the respective age group?
2) In addition, it is essential to distinguish between the different sports of track and field, as they differ greatly in the requirement profile and in the extent of the RAE.
3) In terms of statistics, the results must be presented with odds ratios and confidence interval
Author Response

(The authors gave the same response as above.)

Round 2
Reviewer 1 Report
I am happy with the current version of the manuscript.
The authors did a good job on reviewing the manuscript and answering all the revisions maded.
However, I think that moderate English changes are required.
Author Response
Thank you for your professional suggestion and kind regards to the article.
We have made the necessary spelling and stylistic changes in the text.
Now text looks much better and we hope that it fulfils your expectations.
Authors
Reviewer 2 Report
Thanks to authors for considering my comments. Some improvements were observed. There is still some points that need to be adressed.
In general, I think that mecahnisms underlying RAE in sport can be explained. Excellent sources from D. Hancock should be consulted.
https://www.frontiersin.org/research-topics/11676/birth-advantages-and-relative-age-effects-exploring-organisational-structures-in-youth-sport#articles
1) Some minor spelling issues are still present. Line #94: why young ? is a word lacking ?
2) Reporting results should be improved: example: chi-sq should be presented before p value (see line # 153, 160...). Please revise extensively.
3) In results: Does the Chi square shouldn't be presented with the greek symbol ?
4) I still think that the transitional phase beteween the Soviet years (in comparisons) to Russian system should be adressed. If authors feel uncomfortable to do the analyses they at least should treat this point to a potential limitation, in which they could inform the reader more extensively. As they mention in point 5 (first review), this is important for readers.
As authors suggest, I am really curious to see if RAE was different ath this period. This could be done by conducting analysis of monotonous variations (kind of repeated chi-sq analysis).
5) The additional references are adeuate and relevant to the field.
Author Response
We fully agree with your comments and supplemented the discussion with data from D. Hancock's study.
We've also made all the other changes you mentioned.
Unfortunately, as we pointed out earlier, we do not have data on the dates of birth of the best young athletes participating in the “Shipovka Yunykh” in the USSR.
Although the comparison of the prevalence of RAE in the USSR and modern Russia is really of great interest.
Thank you for you specific and professional suggestions.
Authors
Reviewer 3 Report
The present study examined the prevalence of the Relative Age Effect (RAE) in the best young (10 to 15 years) track and field athletes.
Despite the interesting topic and the lage data set, I strongly suggest the following aspects to improve the quality of the manuscript.
1) please include ORs in the abstract.
2) please seperate the 3 differents events (60m, 600m, long jump). From my point of view long jump and 600m run are very different in key performance indicators (and possibly in RAEs as well).
3) please provide ORs for all the results (female vs. male; events; age groups)
4) optionally you might discuss solutions how RAEs can be minimised.
Author Response
Thank you for you specific and professional suggestions. We tried to fully reflect the answers to your comments.
Regarding the division into groups by disciplines, we can report that all participants in the study performed in all three disciplines (60 meters, 600 meters and jumps).That is, their best results in the all-around were analyzed.
Also, unfortunately, we do not have protocols with the results of competitions in individual disciplines and took into account only the final results of the all-around.
We have indicated this limitation in the relevant section of the article.
As tables in these comparisons were larger than 2x2 we provided Cramer`s V coef. If needed we can change the results so that it will be possible to make 2x2 comparisons, however, it seems to us that this will not fundamentally affect the results obtained.